# Bacterial community shifts of commercial apples, oranges, and peaches at different harvest points across multiple growing seasons

Madison Goforth[1], Margarethe A. Cooper[1], Andrew S. Oliver[2], Janneth Pinzon[3], Mariya Skots[3], Victoria Obergh[1], Trevor V. Suslow[3], Gilberto E. Flores[4], Steven Huynh[5], Craig T. Parker[5], Rachel Mackelprang[4], Kerry K. Cooper[1,6]*

1 School of Animal and Comparative Biomedical Sciences, The University of Arizona, Tucson, Arizona, United States of America, 2 USDA-ARS Western Human Nutrition Research Center, Davis, California, United States of America, 3 Department of Plant Sciences, University of California, Davis, Davis, California, United States of America, 4 Department of Biology, California State University, Northridge, Northridge, California, United States of America, 5 Produce Safety and Microbiology Research Unit, Western Regional Research Center, Agricultural Research Service, USDA, Albany, California, United States of America, 6 BIO5 Institute, University of Arizona, Tucson, Arizona, United States of America

* kcooper@arizona.edu

**Data Availability Statement:** All sequence reads generated for this study are available through the NCBI's SRA archive under the Accession numbers:

## Abstract

Assessing the microbes present on tree fruit carpospheres as the fruit enters postharvest processing could have useful applications, as these microbes could have a major influence on spoilage, food safety, verification of packing process controls, or other aspects of processing. The goal of this study was to establish a baseline profile of bacterial communities associated with apple (pome fruit), peach (stone fruit), and Navel orange (citrus fruit) at harvest. We found that commercial peaches had the greatest bacterial richness followed by oranges then apples. Time of harvest significantly changed bacterial diversity in oranges and peaches, but not apples. Shifts in diversity varied by fruit type, where 70% of the variability in beta diversity on the apple carposphere was driven by the gain and loss of species (i.e., nestedness). The peach and orange carposphere bacterial community shifts were driven by nearly an even split between turnover (species replacement) and nestedness. We identified a small core microbiome for apples across and between growing seasons that included only *Methylobacteriaceae* and *Sphingomonadaceae* among the samples, while peaches had a larger core microbiome composed of five bacterial families: *Bacillaceae*, *Geodermtophilaceae*, *Nocardioidaceae*, *Micrococcaeceae*, and *Trueperaceae*. There was a relatively diverse core microbiome for oranges that shared all the families present on apples and peaches, except for *Trueperaceae*, but also included an additional nine bacterial families not shared including *Oxalobacteraceae*, *Cytophagaceae*, and *Comamonadaceae*. Overall, our findings illustrate the important temporal dynamics of bacterial communities found on major commercial tree fruit, but also the core bacterial families that constantly remain with both implications being important entering postharvest packing and processing.

SRX20084534 (https://www.ncbi.nlm.nih.gov/sra/?term=SRX20084534), SRX20084535 (https://www.ncbi.nlm.nih.gov/sra/?term=SRX20084535), SRX20084536 (https://www.ncbi.nlm.nih.gov/sra/?term=SRX20084536), SRX20084537 (https://www.ncbi.nlm.nih.gov/sra/?term=SRX20084537), SRX20084538 (https://www.ncbi.nlm.nih.gov/sra/?term=SRX20084538), and SRX20084539 (https://www.ncbi.nlm.nih.gov/sra/?term=SRX20084539), and also associated with BioProject Accession number: PRJNA957757 (https://www.ncbi.nlm.nih.gov/bioproject/?term=PRJNA957757).

**Funding:** Funding for the study was provided by the United States Department of Agriculture (USDA), National Institute of Food and Agriculture (NIFA) Award #2017-67018-26173 awarded to Kerry Cooper and Trevor Suslow, and Technology and Research Initiative Fund (TRIF) provided to Kerry Cooper by the University of Arizona. No funding agency had any role in the study design, data collection and analysis, decision to publish, or preparation of the manuscript. USDA NIFA (https://www.nifa.usda.gov/) and UArizona TRIF (https://research.arizona.edu/trif#:~:text=Through%20TRIF%2C%20or%20the%20Technology,largest%20economic%20engines%20for%20Arizona).

**Competing interests:** The authors have declared that no competing interests exist.

## Introduction

Microorganisms interact with plants at several spatial levels in soil and aerial spheres of influence. True epiphytes and endophytes are key interacting components, at the closest level, of fruit and vegetable microbiomes [1, 2]. Recent studies utilizing high-throughput sequencing methods, such as 16S rRNA gene sequencing and shotgun metagenomics, have begun to characterize the microbiomes associated with tree fruit production and handling [2, 3] including profiles of the orchard soil [3], rhizosphere [4], bark [5], and phyllosphere [7] communities. Importantly, the phylogenetic diversity of microbial communities associated with anatomical parts of fruit-bearing trees and throughout orchards varies substantially [1]. Many may be transient non-replicating populations from environmental deposition while a more limited subset is capable of sustained net growth. However, except for apples [8, 8], little is known about the carposphere microbiome, particularly at the point of harvest. Crucially, bacterial communities of the tree fruit carposphere impact storage life, optimize postharvest process controls, and provide novel approaches to food safety systems and management.

Pome fruit, which includes apples and pears, rank among the most consumed fruits globally. These fruits are frequently held under long-term refrigeration to allow for sustained and stable postharvest marketing [8, 8]. The microbiomes of apples, individual apple trees, and orchards have been studied in several different countries [8–10], while including distinct aspects of the rootstock [12], rhizosphere [4, 12], soils [3, 13–15], bark [5, 17], floral nectar [18], and phyllosphere [7]. Moreover, others have investigated the carposphere microbiome of apples during a single harvest time point [8, 8, 18], under several different postharvest conditions, in addition to comparisons between conventional with organic crop management systems [9, 10, 19–21]. Multiple studies demonstrate variation in apple bacterial diversity across the carposphere, production and management strategies, geography, and post-harvest practices. Organic Arlet apples have increased bacterial alpha diversity within or on the stem, stem-end, fruit pulp, and seeds compared to conventionally managed apples [8, 8]. Similarly, alpha diversity of fungal communities in Red Delicious apples was increased on organic fruit as compared to conventional sources [1, 8, 18]. However, variability in the structure of the microbiome was found to be dependent, in part, on geographic locale, which has an outsized role in structuring fungal communities [8]. Beyond orchard management and geographic locale, postharvest processes can impact the apple microbiome. For example, storage conditions can result in substantial modifications to the apple microbiome [9]. Similarly, Abdelfattah and colleagues found that washing and waxing significantly influenced the diversity and composition of the apple microbiome [19]. A recent study by Zhimo et al, found that the carposphere microbiome of apples was influenced by fruit genotype, developmental stage, and storage times. They also found that carposphere microbial communities changed during the developmental process [22]. However, to date no study has longitudinally sampled directly from the apple orchard at different harvest time points during a growing season or between multiple seasons.

Few studies have investigated the composition of the carposphere microbiome for citrus. Much of the existing work has focused on the rhizosphere and phyllosphere. Due to the ubiquitous consumption of citrus, such as oranges, knowledge of their microbiomes could have wide-spread implications for quality and safety [23–26]. Like apples, the bacterial communities found on the phyllosphere of oranges, varies based on geographic locale and management inputs and practices. A study conducted in South Africa found that postharvest practices altered bacterial and fungal diversity and abundance on citrus carpospheres, which parallels results from apple research [24]. Even less has been reported for peaches; the only study of the peach carposphere focused on mummified peaches as a major source of fungal disease [26].

All other microbiome studies involving peach orchards have examined the microbiome of phloem tissues [27] or roots/rootstocks [29]. Like citrus, very little is known about the bacterial diversity of the peach carposphere at harvest maturity let alone temporal changes of the microbiome during growing seasons.

The overall goal of this study was to determine bacterial community structure of carpospheres at the point of harvest for major types of commercial tree fruit—pome (apples), stone (peaches), and citrus (oranges) fruit. Specifically, the study addresses how the bacterial communities present on different types of tree fruit change over time. A strength of this study is that these fruits were sampled directly from the commercial orchards, coincident with harvest maturity, during consecutive growing seasons. The analyses included characterization of the viable and total bacterial communities present on each type of tree fruit at the point of harvest in a commercial operation. Although the overall community composition varied significantly between the three types of tree fruit, we still established shifts in microbial communities occurred during the growing season and between seasons for each of the tree fruit types. These results will help address a knowledge gap in the ecology of the bacterial communities of carpospheres from the fruit environment, and expansion of these microbiome profiles may lead to novel tools and solutions in quality and safety management. For example, understanding the core microbiome of these different tree fruit at the point of harvest can be utilized to develop microbial indexes for verifying and validating sanitizer wash water treatment systems to address requirements of the Produce Safety Rule of the Food Safety and Modernization Act (FSMA).

## Materials and methods

### Tree fruit sample collection

The total number of each type of tree fruit collected at the different harvest time points in the growing season are different among the two growing seasons due to limitations of access to the commercial orchards. At each sampling point, approximately 50 fruits were collected from the same tree and combined into composites of either 10 apples, 10 oranges or 5 peaches. Fruit was harvested at positions between 1.5 and 2.5 m from the orchard floor and included locations on both sides of a row. In season 1, there were 440 total apples from 9 trees, while in season two there were 170 total apples from 4 trees. For oranges, there were 610 total collected from 13 trees in season 1 and 840 total collected from 17 trees in season 2. Lastly, the total collection for peaches was 400 from 8 trees in season 1 and 600 from 12 trees in season 2. The total number of composite samples of the fruit that were collected, processed, and sequenced for the study are listed in Table 1, however some were eliminated during rarefication due to low sequence numbers as described below. Fruit was collected directly from trees at each time

**Table 1. Total composite tree fruit samples collected during study.**

|  | Fruit type | Early Harvest (Total) | Early Harvest (PMA) | Middle Harvest (Total) | Middle Harvest (PMA) | Late Harvest (Total) | Late Harvest (PMA) | Total |
|---|---|---|---|---|---|---|---|---|
| Season 1 | Apples | 12 | 7 | 7 | 6 | 6 | 6 | 44 |
|  | Peaches | 23 | 18 | 0 | 0 | 20 | 19 | 80 |
|  | Oranges | 0 | 0 | 27 | 22 | 6 | 6 | 61 |
| Season 2 | Apples | 9 | 8 | 0 | 0 | 0 | 0 | 17 |
|  | Peaches | 40 | 40 | 0 | 0 | 24 | 16 | 120 |
|  | Oranges | 24 | 23 | 0 | 0 | 21 | 16 | 84 |
|  | Total | 108 | 96 | 34 | 28 | 77 | 63 | 406 |

point of harvest from commercial orchards in Courtland, CA, Exeter, CA, or Kingsburg, CA, for apple cultivar Granny smith, Thompson Improved Navel oranges, and peaches, respectively. Commercial orchards in this context are orchards that have the intent on selling and distributing produce for public consumerism. These specific orchard locations were chosen because of the agreement and accessibility with the commercial growers to participate in the study. At the point of collection, the fruits were placed in sterile plastic bags, then placed on ice in coolers for transport back to the laboratory. Samples were processed immediately upon arrival at the laboratory as described below. The study was conducted across two growing seasons (2017 and 2018), each type of fruit was sampled at 1–3 different time points throughout commercial growing season in California (Sampling time range for study–(1) Peaches: June—September, (2) Oranges: March—April, (3) Apples: August to October). Only fruit from trees that were coincident with commercial harvest activity were selected at a given harvest time point within the different sampling time points during the production season. No permits were required for this field work, as these were commercial orchards and permission to access these orchards were provided by the owner/operator/grower of each orchard.

## Sample processing

Each composite sample was prepared by placing one fruit in 10 ml sterile detergent solution (0.3 M sodium chloride, 0.1% Tween 20) inside a sterile bag that was then hand massaged for 1 minute, the fruit was aseptically removed. The next fruit was washed in the bag until all fruits for the composite had been washed. Next, the fruit rinsate from each composite sample was centrifuged (10,000 x g; 15 mins), the supernatant discarded, and the pellet resuspended in 100 μL phosphate buffered saline (PBS) for DNA extraction, except peach rinsates were strained through sterile cheese cloth (as determined to be necessary during preliminary studies) to largely exclude peach fuzz from the rinsate prior to centrifugation. Approximately half the composite samples from the same tree were used to determine the viable bacterial communities or microbiome as previously described (Vaishampayan et al., 2013) using propidium monoazide (PMA; Biotium, Fremont, CA, USA) prior to DNA extraction [29]. Modifications were made to the protocol based on preliminary experiments. Briefly, PMA was added to the pellet to generate a final concentration of 50 μM, vortexed for 3 mins, incubated for 50 mins including inverting the tubes 10 times every 10 mins, and then placed in PMA Lite device (Biotium) for 20 mins. All PMA sample processing was done in the dark due to the light sensitivity of the PMA, and after PMA treatment samples were used for DNA extraction.

## DNA extraction

Both viable (PMA-treated rinsate) and total (untreated rinsate) composite samples were DNA extracted using the FastDNA Spin Kit for Soil (MP Biomedicals, Irvine, CA, USA) according to the manufacturer's instructions. Extracted DNA was quality checked and quantified using a Nanodrop spectrophotometer (Thermo Fisher Scientific) prior to 16S rRNA gene PCR amplification.

## 16S ribosomal RNA amplification and sequencing

Extracted DNA from the composite samples during the first growing season were PCR amplified using one of two primer sets, either 515F—926R primers that amplify the V4-V5 regions or 799F—1115R primers that amplify the V5-V6 regions of the 16S rRNA gene. The 799F—1115R primers were designed to exclude most chloroplast sequences from plant-based samples like tree fruit carpospheres [20, 30, 32]. PCR amplification was conducted by using the following reaction set up: 10 μL of Platinum Hot Start PCR Master Mix (2x; Thermo Fisher), 1 μL of

pre-mixed 515F—926R primers (10 μM concentration) or 1 μL of pre-mixed 799F—1115R primers (10 μM concentration), 9 μL PCR grade water (Qiagen), and 5 μL template DNA (due to low biomass, quantification indicated this amount to get enough DNA for proper amplification) and sterile nuclease-free water served as negative controls. Primers 515—926R PCR amplification was conducted with the following conditions: 95˚C for 3 mins, 30 cycles: 95˚C for 45 secs, 50 ˚C for 45 secs, and 68 ˚C for 90 secs, and finally 68 ˚C for 5 mins. Primers 799F —1115R PCR amplification was conducted with the following conditions: 94˚C for 3 mins, 35 cycles: 94˚C for 45 secs, 54 ˚C for 60 secs, and 72 ˚C for 90 secs, and finally 72 ˚C for 10 mins. To determine successful amplification post-PCR, products were visualized using 1.5% agarose gel. Negative water controls showed no bands on the gel.

All amplicons were quantified using Quant-iT PicoGreen dsDNA Assay (Invitrogen) per the manufacturer's instructions. After quantification, all samples were pooled together in equal molar ratios for sequencing. Pooled barcoded amplicon libraries were cleaned using QIAquick PCR Purification Kit (Qiagen) according to the manufacturer's instructions. Final libraries prepared for sequencing were quantified with a Qubit 4.0 fluorometer and then sequenced on an Illumina MiSeq sequencer at California State University, Northridge (CSUN) using MiSeq reagent kit v2 (300-cycles, Illumina) for 150 bp reads or at the Produce Safety and Microbiology Section, Agricultural Research Service, United States Department of Agriculture using MiSeq reagent kit v3 (600-cycles, Illumina) for 300 bp reads.

## Sequencing read processing

Only sequence reads generated using both the same primer set (515F-926R or 799F-1115R) and MiSeq reagent kit (300 cycles or 600 cycles) were combined, processed, and analyzed together (e.g., 515F-926R primers– 300 cycles, 799-1115R primers– 600 cycles, etc.). All sequence reads were demultiplexed, quality trimmed, and merged using the QIIME2 software (v2020.2) [32]. Briefly, sequence reads were demultiplexed. The average quality of each individual base in the reads was assessed reads with quality scores below <Q35 were removed. Additionally, for the 799F—1115R forward reads that were sequenced using the barcode sequencing primer the first 42 base pairs were also removed as they represented part of the primer and barcode sequence not part of the 16S rRNA gene. Reads that were 150 bp in length only used the reverse reads for further analysis. Forward and reverse reads from 300 bp sequence runs were trimmed, denoised and merged using the DADA2 plugin [33] in QIIME2. Reads from different runs with the same primer and sequence length were processed independently and then merged in QIIME2. After initial comparison of the different primers and read length of the samples, all remaining data analysis in this study utilized the 300 bp reads generated with the 799F—1115R primers. All sequence reads generated for this study are available through the NCBI's SRA archive under the Accession numbers: SRX20084534, SRX20084535, SRX20084536, SRX20084537, SRX20084538, and SRX20084539, and also associated with Bio-Project Accession number: PRJNA957757.

## Taxonomic classification

Taxonomic assignment for the 799F—1115R primers (chloroplast excluding) was done using the feature-classifier plugin in QIIME2 with the Greengenes database (v13.8) with 99% sequence similarity. The classifier was trained on the 799F primer 5′ – AACMGGATTAGATACCKG–3′ and 1115R primer 5′ – AGGGTTGCGCTCGTTG–3′ with a minimum length of 200 bp and maximum length of 500 bp.

## Alpha and beta diversity analysis

The OTU table file and the tree file were generated using QIIME2 commands for creating the phylogenetic unrooted tree file and taxonomy file. Both were exported with the additional commands that formatted the unrooted file as the tree file and the taxonomy file as the OTU table file. These were formatted and imported into R for further analysis using the packages: (1) phyloseq (v.1.40.0) [34], (2) microbiome (v.1.18.0) [35, 37], and (3) vegan (v.2.6.2) [38]. After importing a phyloseq object was created from the files and a metadata sheet. All samples were filtered to eliminate any chloroplast and mitochondria sequences that were amplified using the 799F—1115R primer set. Samples were rarefied to an even sequencing depth of 2000 sequences per sample using phyloseq. Alpha diversity was calculated using the Shannon index, Chao index, and species richness with phyloseq and microbiome. Pairwise Kruskal-Wallis tests were used to compare alpha diversity indices and pairwise Wilcox tests were used for comparisons of Shannon diversity against metadata variables (i.e., total vs viable, growing season) from the stats package. Beta diversity was investigated using both weighted and unweighted Unifrac and Bray-Curtis dissimilarity distance matrices with phyloseq and microbiome. Compositional variability was calculated using adonis2 against Bray-Curtis distances with 999 permutations from the vegan package (v.2.6.2).

## Taxonomic composition visualization and core microbiome analysis

Taxonomic analysis was conducted using the phyloseq, microbiome, microbiomeutilities (v.1.0.16) [34, 37], and vegan packages [38] in R. Taxonomic visualization of bacterial families found in the fruit samples at different harvest points or growing seasons were generated with a 1% relative abundance and 75% prevalence after rarefaction of the data. Viable (PMA-treated) samples were analyzed separately from total (PMA-treated and non-PMA-treated) samples where appropriate. Core microbiome analysis of the different fruit samples was also conducted using the phyloseq package in R. Tree fruit samples were analyzed as a whole, and then filtered by type of fruit for core members. Bacteria were considered core if they occurred at a relative abundance of $\geq 0.001$ and prevalence of at least 75% of the target samples [8]. An additional core analysis more relaxed filters of at least 95% of the target samples with abundance of $\geq 10e^{-10}$ was also conducted.

## Additional data analysis

Dissimilarity matrices for viable and total tree fruit microbiomes were computed using the beta-dispersion command under vegan and betapart (v.1.5.6) [39] packages. Beta-dispersion was tested against harvest time points as Euclidean distances for Sorensen dissimilarity looking at the presence or absence of species; Simpson dissimilarity looking at the replacement or turnover of species; SNE dissimilarity looking at the nestedness of species (Sorensen—Simpson); and Beta-total looking at compositional variance of species (SNE + Simpson). Mantel tests were done from the vegan package utilizing the beta-dispersion dissimilarities distance matrix against the euclidean temporal distance to assess varying species dissimilarities against the temporal correlation (i.e., varying harvest points). Top bacterial taxa biomarkers for both viable and tree fruit samples were analyzed with the linear discriminate analysis (LDA) effect size (LefSe) using the lefser (v.1.6.0) (Khleborodova 2022) and microbiomeMarker (v.1.3.2) packages [40, 41]. We used an LDA cutoff of 4, which was stricter than default [22].

## Results

To determine the composition of bacterial communities and to measure bacterial diversity of tree fruit samples, we amplified and sequenced variable regions of the bacterial 16S RNA gene. Initially, we investigated whether using primers targeting different variable regions (i.e., V4-V5, V5-V6) would lead to variation in measured bacterial diversity or compositional differences. While there were shifts in the identified bacterial diversity in the same samples using different primers and read lengths there were not any statistically significant differences in the diversity (S1 Fig). Therefore, the remaining analyses utilized 300 bp paired-end reads generated using the chloroplast excluding 799F—1115R reads.

We next asked whether there were differences in alpha diversity between different fruit types. Commercial peaches were found to have the greatest level of bacterial richness (22,768 observed ASVs) followed by commercial oranges (20,296 observed ASVs) and then commercial apples (4,780 observed ASVs). Using several different alpha diversity indices, we found significant pairwise differences between apples and peaches, oranges and peaches, but not for apples and oranges (S2 Fig). Similar alpha diversity results were obtained when we analyzed only the viable bacterial communities of the carposphere (S3 Fig). We next asked whether there were differences in bacterial composition between different fruit types. We found that overall, fruit type explained a significant amount of variation in bacterial composition (PERMANOVA, p = 0.001, $R^2$ = 0.16). We also found a significant temporal component to bacterial composition. Within each fruit type, the bacterial community composition (total or viable only) varied significantly based on the point of harvest in the growing season and the specific season of harvest (PERMANOVA, p = 0.001, $R^2$ = 0.22, 0.30, 0.26) (S4 Fig). Additionally, we visualized the taxonomic changes underpinning differences in alpha and beta diversity using bar plots (S5 Fig). Having established significant differences in the bacterial communities between apples, oranges, and peaches, we next applied more detailed subsequent analyses on the individual fruits themselves.

## Apples

While we found subtle differences in the diversity metrics of apples depending on the point sampled during the growing season, none of these changes were statistically significant. When we assessed alpha diversity using the Shannon Diversity index, there were no statistically significant results amongst either growing season (p-value > 0.05) nor the seasonal variation (p-value > 0.05) for apples. For example there was no significant difference in bacterial communities between the total samples harvested late in the first season and viable samples harvested early in the second season for apples (Fig 1A; p-value = 1.00), total bacterial communities compared to the viable bacterial communities for early in the first season (p-value = 1.00), or the viable bacterial communities between early 1st season and the middle of the 1st season (p-value = 0.22). Overall, the harvest points for total and/or viable bacterial communities on apples were not significantly altered during the growing season (Fig 1A; p-value > 0.05).

Diversity of the apple microbiome based on the Bray-Curtis dissimilarity distance matrix indicates there were shifts in the bacterial communities at each of the harvest time points for each season and between the two seasons based on the clustering of the samples. Each of the viable bacterial community samples clustered with the corresponding harvest time point for total bacterial community samples, usually tightly except the 1st season late harvest samples that were more spread out from the other total bacterial communities. Both the total and viable samples from early harvest apples during the 1st season clustered closely to the early in the 2nd growing season apple samples, which indicates there may be a cyclic nature to the diversity of the bacterial communities on apples in the orchards based on the Bray-Curtis PCoA plot

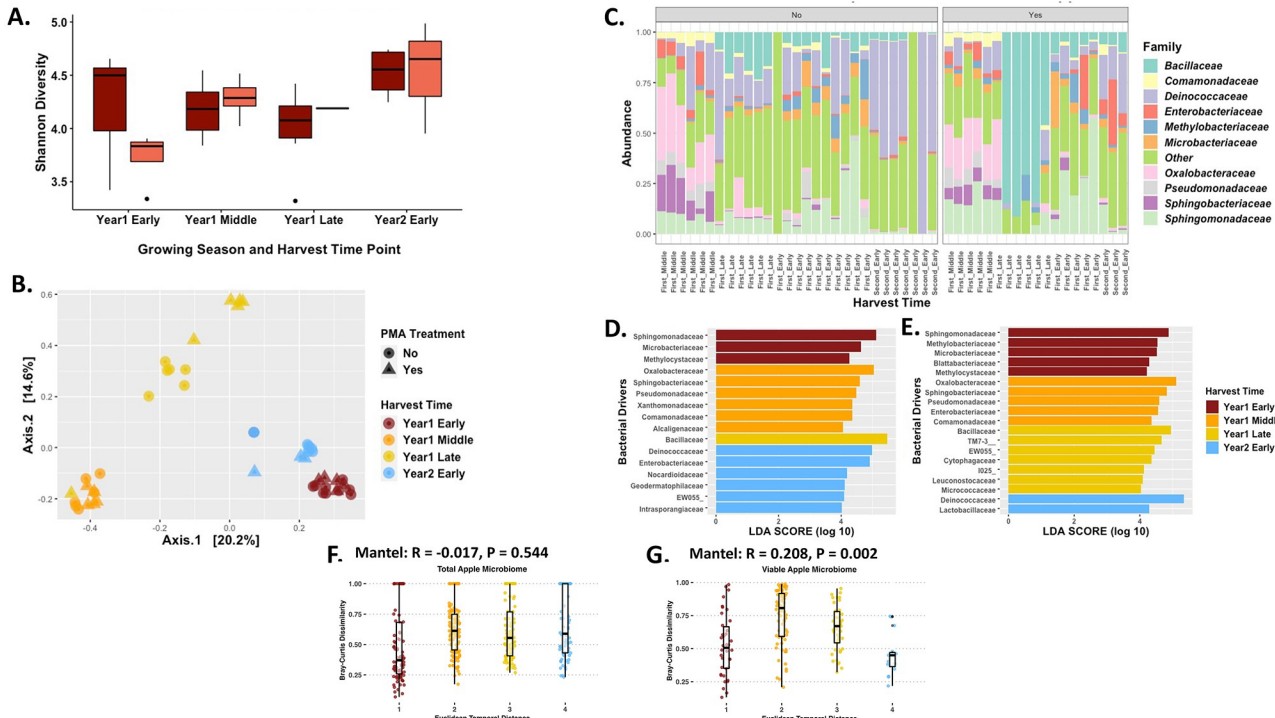

**Fig 1. Bacterial diversity of the apple carposphere.** (A) Shannon Diversity Index for apples across different time points in the growing season across two growing seasons. Dark red boxes represent total apple carposphere samples, while the light red boxes are the viable apple carposphere samples. Kruskal-Wallis and Wilcox analysis showed no significance for diversity of apple microbiome over harvesting times (p-value > 0.01). The Kruskal-Wallis p-value was 0.09 for harvest time periods and p-value of 0.45 for the treatment. Wilcox pairwise analysis among the harvest times had a single point at p-value = 0.44 and the rest at p-value = 1.00. (B) PCoA plot based on the Bray-Curtis Dissimilarity Distance Matrix of all the apple carposphere samples. (Permutations = 999; $R^2$ = 0.26; p-value = 0.001). (C) Relative abundance of the top 10 taxa that are present on the different apple carposphere samples, group clustered under "No" represent total microbiome samples that were not treated with PMA, whereas the group clustered under "Yes" represent the viable microbiome samples that were treated with PMA prior to DNA extraction. (D) Linear discriminant analysis Effect Size (LEfSe) analysis of total apple carposphere samples determine those taxa that were most likely driving the differences between the different harvest time points. (E) Linear discriminant analysis Effect Size (LEfSe) analysis of viable apple carposphere samples determine those taxa that were most likely driving the differences between the different harvest time points. (F) Mantel test using the Bray-Curtis Dissimilarity Distance Matrix against the Euclidean temporal distance assesses the temporal correlation of the taxa abundance for total bacterial communities of the apple carposphere. (G) Mantel test using the Bray-Curtis Dissimilarity Distance Matrix against the Euclidean temporal distance assesses the temporal correlation of the taxa abundance for viable bacterial communities of the apple carposphere. Euclidean temporal distance x-axis labels are numerical and represent the harvest time points. In sequential order, 1 is first season early harvest, 2 is first season middle harvest, 3 is first season late harvest, 4 is second season early harvest, and 5 is second season late harvest.

(Permanova 999; p-value = 0.001, $R^2$ = 0.26; Fig 1B). Unfortunately, we were not able to obtain additional samples later in the $2^{nd}$ growing season, so additional research across multiple seasons is needed to confirm if there is a cyclic nature to the bacterial diversity on the apple carposphere.

We further investigated variation in microbial community structure by identifying and characterizing highly abundant bacterial families that change during and between seasons. Early in the $1^{st}$ season, *Sphingomonadaceae*, *Deinococcaceae*, and *Bacillaceae* were prevalent families, but decreased by the middle of the season. These middle season decreases were countered by an increase in *Sphingobacteriaceae* and *Pseudomonadaceae* before an increase in the original three families (*Sphingomonadaceae*, *Deinococcaceae*, and *Bacillaceae*) late in the growing season. Additionally, levels of *Enterobacteriaceae* were found to be higher in more viable samples compared to the total microbiome samples (Fig 1C). Prevalence of these families were

further supported when examining taxonomic drivers of the bacterial communities for each of the harvest time points. However, there were some families that were taxa drivers but were not highly abundant, like *Methylocystaceae* early in the 1st season or *Oxalobacteraceae* in the middle of the 1st year. *Enterobacteriaceae* was a major taxa driver early in the 2nd year right behind *Deinococceae* (Fig 1D). Examination of the taxa drivers in the viable microbiome had some commonalities compared to the total microbiome at certain time points, but also some differences like *Blattabacteriaceae* early in the 1st season or *Enterobacteriaceae* in the middle of the 2nd season. Interestingly, there were some major shifts in the number of bacterial taxa drivers between viable versus total bacterial communities, such as only *Bacillaceae* late in the 1st year versus seven different taxa drivers late in the 1st season time point for the viable microbiome. In contrast, there were six bacterial taxa drivers in the total microbiome early in the 2nd season, but only two in the viable bacterial communities (Fig 1E).

To better understand the temporal correlation on the abundance of taxa we conducted Mantel tests on the Bray-Curtis dissimilarity matrix versus the Euclidean temporal distance matrix. The Mantel test showed a negative correlation and no significant dissimilarity relationship for viable apple samples (Fig 1F), but a positive correlation and significant relationship for total apple samples (Fig 1G). Next, to further characterize the temporal correlation (harvest time point) against bacterial variance for viable and total apple microbiome over time, we looked at taxa presence or absence (β-Sorensen Pairwise Dissimilarity), species turnover (β-Simpson Pairwise Dissimilarity), and species nestedness (β-SNE Pairwise Dissimilarity (Sorensen—Simpson)) with Mantel tests.

The Sorensen Mantel tests showed community composition of the total apple samples co-varied significantly with temporal distance, which is highlighted by a positive correlation and a strong dissimilarity relationship as samples become more dissimilar at different points of harvest (Fig 2A). They also become more dissimilar for microbial composition, while there was no correlation and no dissimilarity relationship for viable apples (Fig 2B). The Simpson Mantel test showed a weak positive correlation but not a strong dissimilarity relationship for total apple samples when looking at species replacement over the harvest time period (Fig 2C), while the Mantel test showed a negative correlation and no dissimilarity relationship for viable apple samples (Fig 2D).

The SNE Mantel test showed community composition of total apple samples co-varied significantly with temporal distance, which is highlighted by a positive correlation and a strong dissimilarity relationship (Fig 2E). Whereas there was a weak positive correlation and a weak dissimilarity relationship for viable apple samples (Fig 2F).

Overall, the study found that nestedness was responsible for 71% of the temporal changes for the total bacterial communities, whereas it was 57% for nestedness and 43% for turnover for the viable bacterial communities (Fig 2G). Although there are shifts in the bacterial families between harvest time points and viable versus total microbiomes, there is still a small core microbiome on apples that is present in at least 75% of samples at 0.1% detection. These include *Methylobacteriaceae* and *Sphingomonadaceae* for the total microbiome, but only *Methylobacteriaceae* for the viable microbiome (Table 2). However, both these bacterial families were eliminated if including 95% of the samples regardless of abundancy cut-off levels.

## Oranges

Unlike apples, there were significant shifts in the bacterial diversity of oranges at nearly every harvest time point and between the two growing seasons (p-value< 0.01; Fig 3A). Diversity between the 1st and 2nd late season harvest time points was the only non-significant comparison. Also, like apples, there were no significant differences in diversity between the total

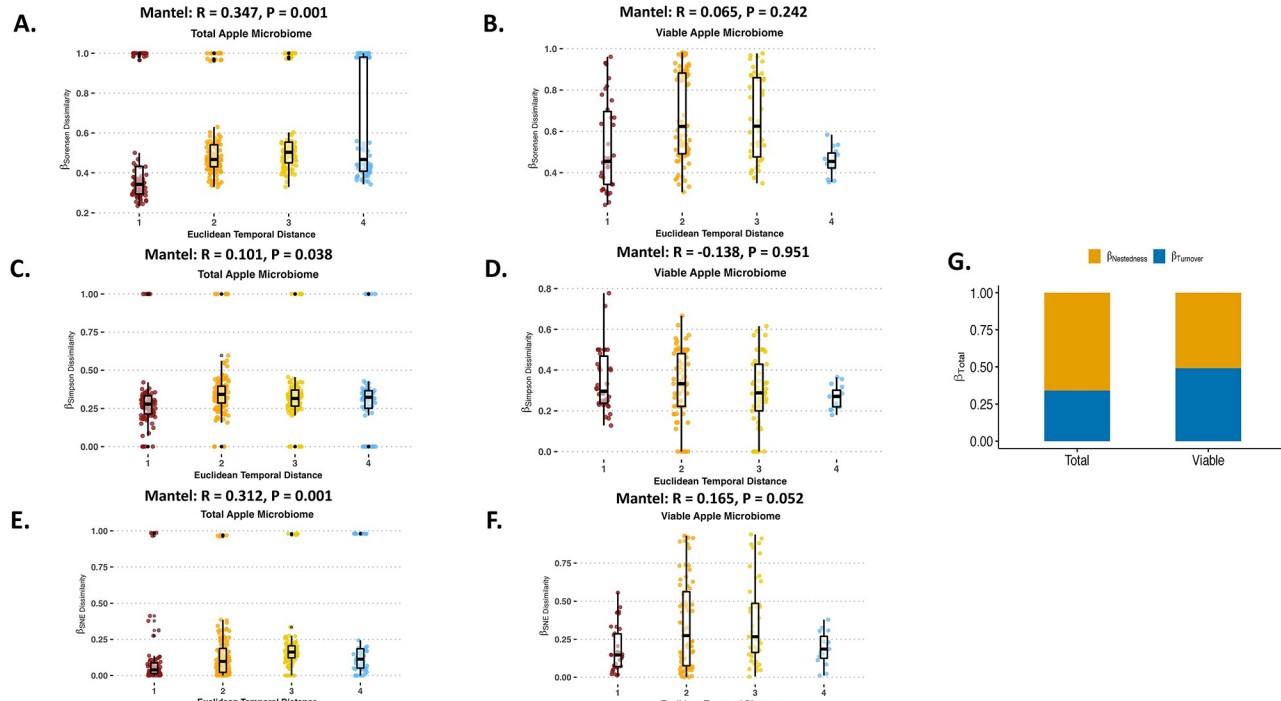

**Fig 2. Role of temporal distance (harvest time point) against bacterial variance for apple carposphere microbiome.** (A) Mantel test using β-Sorensen Pairwise Dissimilarity against the Euclidean temporal distance assesses the temporal correlational role of the taxa presence or absence for total bacterial communities of the apple carposphere. (B) Mantel test using β-Sorensen Pairwise Dissimilarity against the Euclidean temporal distance assesses the temporal correlational role of the taxa presence or absence for viable bacterial communities of the apple carposphere. (C) Mantel test using β-Simpson Pairwise Dissimilarity against the Euclidean temporal distance assesses the temporal correlational role of the species replacement or turnover for total bacterial communities of the apple carposphere. (D) Mantel test using β-Simpson Pairwise Dissimilarity against the Euclidean temporal distance assesses the temporal correlational role of the species replacement or turnover for viable bacterial communities of the apple carposphere. (E) Mantel test using β-SNE Pairwise Dissimilarity (Sorensen—Simpson) against the Euclidean temporal distance assesses the temporal correlational role of the nestedness for total bacterial communities of the apple carposphere. (F) Mantel test using β-SNE Pairwise Dissimilarity (Sorensen—Simpson) against the Euclidean temporal distance assesses the temporal correlational role of the nestedness for viable bacterial communities of the apple carposphere. (G) Compositional variance (β-Total) assesses whether species turnover (β-Simpson) or species nestedness (β-SNE) contributes more to viable and total bacterial communities of the apple carposphere. Euclidean temporal distance x-axis labels are numerical and represent the harvest time points. In sequential order, 1 is first season early harvest, 2 is first season middle harvest, 3 is first season late harvest, 4 is second season early harvest, and 5 is second season late harvest.

**Table 2. Core bacterial families of tree fruit[1].**

| | Total Core | | Viable Core | |
|---|---|---|---|---|
| **Apples** | Methylobacteriaceae Sphingomonadaceae | | Methylobacteriaceae | |
| **Peaches** | Bacillaceae Geodermtophilaceae Nocardioidaceae Micrococcaceae Trueperaceae | | Bacillaceae | |
| **Oranges** | Bacillaceae Micrococcaceae Microbacteriaceae Kineosporiaceae Geodermatophilaceae Nocardioidaceae | Methylobacteriaceae Sphingomonadaceae Cytophagaceae Comamonadaceae Oxalobacteraceae Turicibacteraceae | Oxalobacteraceae Cytophagaceae Sphingomonadaceae Methylobacteriaceae Micrococcaceae | Nocardioidaceae Geodermatophilaceae Bacillaceae Turicibacteraceae Microbacteriaceae Comamonadaceae |

[1]Bacterial family present in at least 75% of samples; Abundancy $\geq$ 0.1%

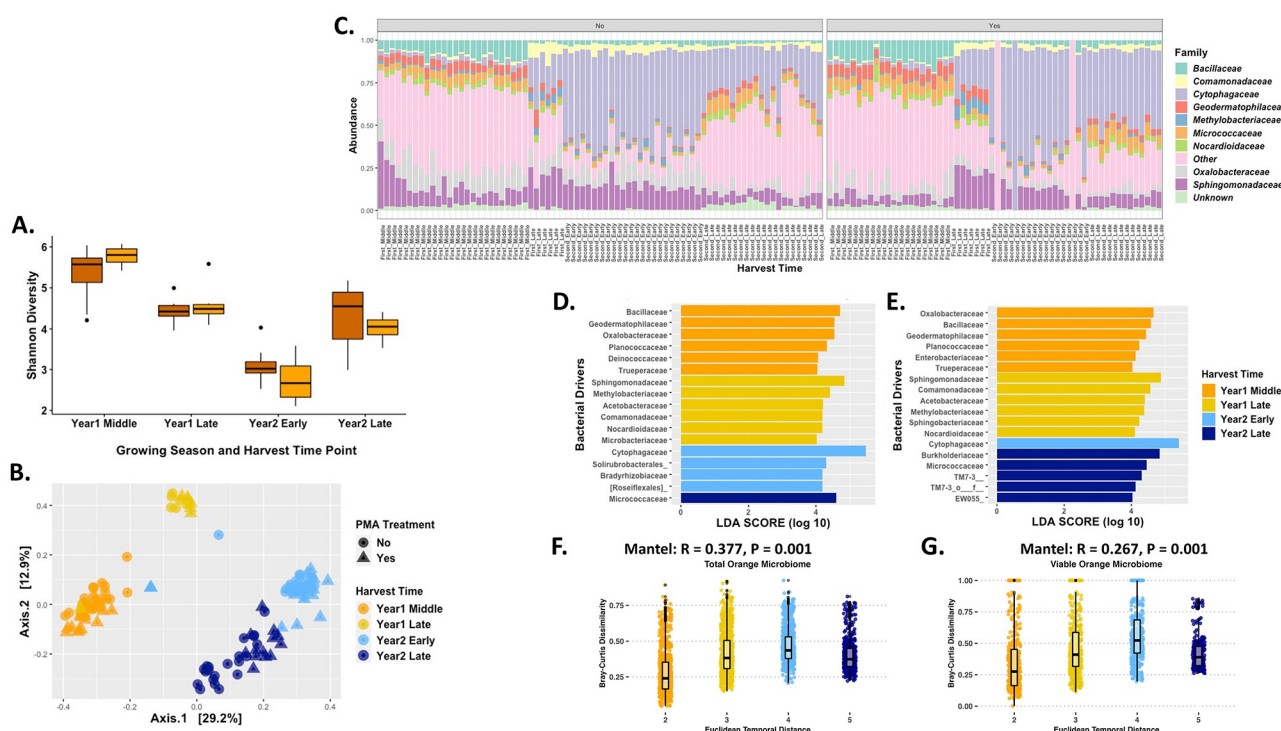

**Fig 3. Bacterial diversity of the orange carposphere.** (A) Shannon Diversity Index for oranges across different time points in the growing season across two growing seasons. Dark orange boxes represent total orange carposphere samples, while the light orange boxes are the viable orange carposphere samples. Kruskal-Wallis and Wilcox analysis showed significance for diversity of orange microbiome over harvesting times (p-value < 0.01), except year one harvest with year two late harvest (p = 0.54). The Kruskal-Wallis p-value was $2.2x10^{-16}$ for the harvesting time periods and p-value of 0.42 for the treatment. Wilcox pairwise analysis among the harvest time periods had significance for different periods. For total early harvest in the first year to total first year middle harvest, total second year early harvest, viable middle harvest first year, and viable second year early harvest, the p-values were 0.014, 0.00019, 0.00019, and 0.0014, respectively. For total middle harvest in the first year to total second year early harvest, total second year late harvest, viable second year early harvest, and viable second year late harvest the p-values were $4.6x10^{-13}$, $4.5x10^{-06}$, $2.4x10^{-09}$, and $1.3x10^{-07}$, respectively. For total early harvest in the second year to total second year late harvest, viable first year late harvest, viable first year middle harvest, and viable second year late harvest, the p-values were $3.4x10^{-07}$, $2.1x10^{-05}$, $1.5x10^{-11}$, and $4.7x10^{-07}$, respectively. For total late harvest in second year to viable first year middle harvest and viable second year early harvest p-values were $1x10^{-10}$ and $9x10^{-06}$, respectively. For viable late harvest in the first year to viable first year middle harvest and viable second year early harvest p-values were 0.00057 and 0.00048, respectively. For viable middle harvest in the first year to viable second year early harvest, viable second year late harvest, the p-values were $2.4x10^{-08}$ and $6.0x10^{-08}$, respectively. Lastly, for the viable early harvest in the second year to viable second year late harvesting had a p-value of $5.6x10^{-06}$. (B) PCoA plot based on the Bray-Curtis Dissimilarity Distance Matrix of all the orange carposphere samples. (Permutations = 999; R2 = 0.29; p-value = 0.001). (C) Relative abundance of the top 10 taxa that are present on the different orange carposphere samples, group clustered under "No" represent total microbiome samples that were not treated with PMA, whereas the group clustered under "Yes" represent the viable microbiome samples that were treated with PMA prior to DNA extraction. (D) Linear discriminant analysis Effect Size (LEfSe) analysis of total orange carposphere samples determine those taxa that were most likely driving the differences between the different harvest time points. (E) Linear discriminant analysis Effect Size (LEfSe) analysis of viable orange carposphere samples determine those taxa that were most likely driving the differences between the different harvest time points. (F) Mantel test using the Bray-Curtis Dissimilarity Distance Matrix against the Euclidean temporal distance assesses the temporal correlation of the taxa abundance for total bacterial communities of the orange carposphere. (G) Mantel test using the Bray-Curtis Dissimilarity Distance Matrix against the Euclidean temporal distance assesses the temporal correlation of the taxa abundance for viable bacterial communities of the orange carposphere. Euclidean temporal distance x-axis labels are numerical and represent the harvest time points. In sequential order, 1 is first season early harvest, 2 is first season middle harvest, 3 is first season late harvest, 4 is second season early harvest, and 5 is second season late harvest.

bacterial communities and the viable at any of the harvest time points through two growing seasons.

Like apples, season and harvest time point significantly affected microbial community structure (Permanova 999; p-value = 0.001; $R^2$ = 0.29; Fig 3B). Additionally, it was consistent with alpha diversity results as the viable microbiome samples clustered tightly with total bacterial community samples. However, unlike apples, there was a clear separation of 1st season

orange samples and 2nd season orange samples, which suggests there may be a weaker cyclic nature to the carposphere microbiome of oranges. However, additional research focused on a potential cyclic nature across numerous seasons would be needed to draw definitive conclusions.

The abundance of the top bacterial families provides some specifics to the seasonal and different harvest time point changes on the orange carposphere. The oranges from the middle of the 1st season had higher levels of *Bacillaceae*, *Geodermatophilaceae*, *Micrococcaceae*, and other bacterial families, which shifted by late in the 1st season and the 2nd season with an increase in *Cytophagaceae* and *Comamonadaceae*. Although by late in the 2nd growing season *Micrococcaceae* and other bacterial families were increasing again on the orange carposphere resulting in decreases in *Cytophagaceae*. The bacterial abundance at the taxonomic level also confirmed no significant differences between the total carposphere microbiome and the viable microbiome except for a higher population of *Methylobacteriaceae* in the viable samples late in the 1st growing season (Fig 3C). A LefSe analysis was conducted to assess those bacterial families that were major taxa drivers of the bacterial diversity at the different harvest time points for oranges. Early in the 1st year *Bacillaceae* was the major taxa driver, but this shifted to *Sphingomonadaceae* by late in the 1st season. During the 2nd growing season *Cytophagaceae* was the major taxa driver early in the season with *Micrococcaceae* being the lone taxa driver late in the 2nd season (Fig 3D). The viable bacterial communities of oranges had similar taxa drivers compared to the total taxa drivers, although viable middle harvest of the 1st year samples had *Enterobacteriaceae* as a taxa driver that was absent from the total samples. Additionally, another difference was the viable bacterial communities early in the 2nd year only had *Cytophagaceae* compared to four taxa drivers for the total samples, whereas the viable samples from late in the 2nd year had five taxa drivers compared to only *Micrococcaceae* in the total samples (Fig 3E).

To understand the temporal correlation on the abundance of taxa of the orange carposphere, we conducted Mantel tests on the Bray-Curtis dissimilarity matrix versus the Euclidean temporal distance matrix, which showed a positive correlation and a strong dissimilarity relationship for both the viable (Fig 3F) and total orange bacterial abundances (Fig 3G). Additional Mantel tests for the taxa presence and absence, turnover, and nestedness for the orange carposphere temporal changes were conducted.

The Sorensen Mantel test showed community composition of the total (Fig 4A) and viable (Fig 4B) orange carposphere samples co-varied significantly with temporal distance, which is highlighted by a positive correlation and a strong dissimilarity relationship as samples become more dissimilar at different points of harvest. The Simpson species replacement Mantel test showed a negative correlation and no dissimilarity relationship for total (Fig 4C) and viable (Fig 4D) orange carposphere samples when looking at the harvest time points. The SNE Mantel test showed community composition of total (Fig 4E) and viable (Fig 4F) orange samples co-varied significantly with temporal distance, which is highlighted by a positive correlation and a strong dissimilarity relationship. Unlike apples, the temporal changes in bacterial communities on oranges for both the total and viable samples were close to even split between nestedness and turnover. For the total microbiome the temporal changes were 55% nestedness versus 45% turnover compared to 59% nestedness to 41% turnover for the viable microbiome (Fig 4G).

The stability of the major bacterial families in the taxonomical profiles between seasons, harvest time points, and viable versus total microbiome suggest there should be numerous bacterial families in the core microbiome of the orange carposphere. In fact, there are 12 bacterial families in the total orange carposphere core microbiome and 11 bacterial families in the viable core microbiome (Table 2). Additionally, the core microbiome for the orange carposphere still

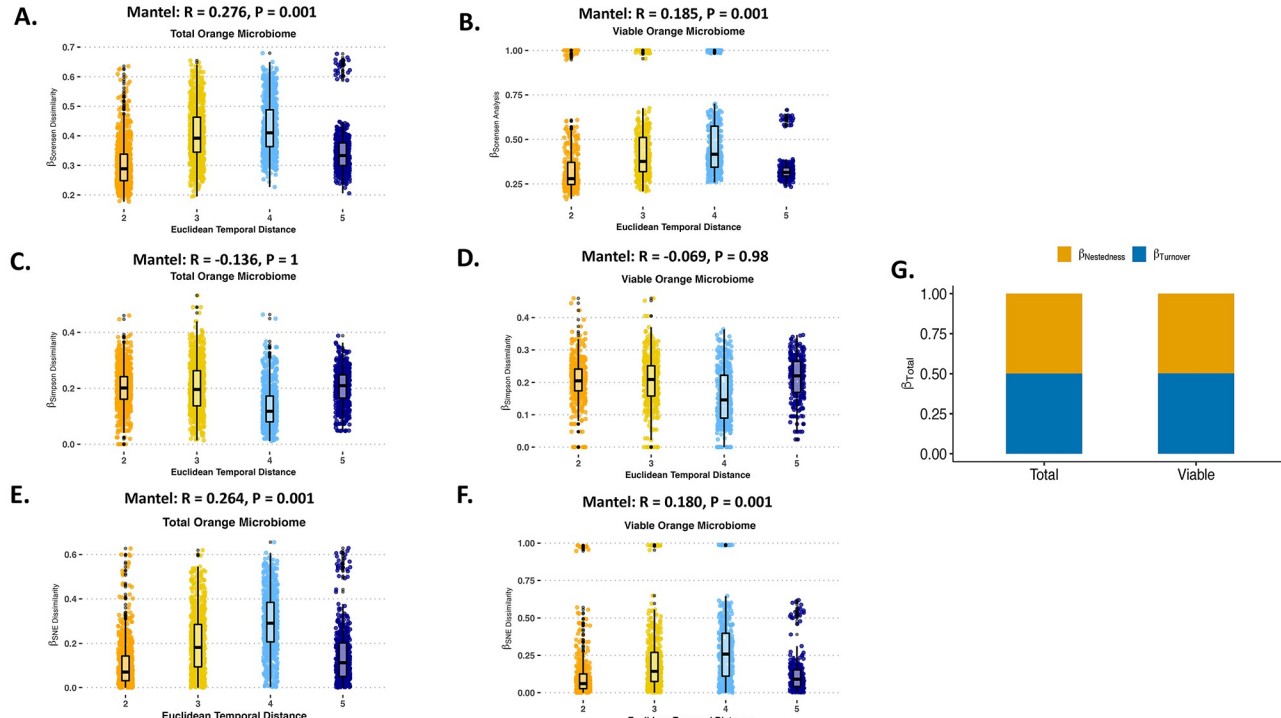

**Fig 4. Role of temporal distance (harvest time point) against bacterial variance for orange carposphere microbiome.** (A) Mantel test using β-Sorensen Pairwise Dissimilarity against the Euclidean temporal distance assesses the temporal correlational role of the taxa presence or absence for total bacterial communities of the orange carposphere. (B) Mantel test using β-Sorensen Pairwise Dissimilarity against the Euclidean temporal distance assesses the temporal correlational role of the taxa presence or absence for viable bacterial communities of the orange carposphere. (C) Mantel test using β-Simpson Pairwise Dissimilarity against the Euclidean temporal distance assesses the temporal correlational role of the species replacement or turnover for total bacterial communities of the orange carposphere. (D) Mantel test using β-Simpson Pairwise Dissimilarity against the Euclidean temporal distance assesses the temporal correlational role of the species replacement or turnover for viable bacterial communities of the orange carposphere. (E) Mantel test using β-SNE Pairwise Dissimilarity (Sorensen—Simpson) against the Euclidean temporal distance assesses the temporal correlational role of the nestedness for total bacterial communities of the orange carposphere. (F) Mantel test using β-SNE Pairwise Dissimilarity (Sorensen—Simpson) against the Euclidean temporal distance assesses the temporal correlational role of the nestedness for viable bacterial communities of the orange carposphere. (G) Compositional variance (β-Total) assesses whether species turnover (β-Simpson) or species nestedness (β-SNE) contributes more to viable and total bacterial communities of the orange carposphere. Euclidean temporal distance x-axis labels are numerical and represent the harvest time points. In sequential order, 1 is first season early harvest, 2 is first season middle harvest, 3 is first season late harvest, 4 is second season early harvest, and 5 is second season late harvest.

included 11 bacterial families when increased to 95% of the samples for the total microbiome but was eliminated in the viable microbiome.

## Peaches

Among the three types of tree fruit, peaches had the greatest level of diversity compared to the other two types of tree fruit, thus the temporal shifts would be expected to be altered slightly compared to apples and oranges. Unlike apples and oranges there were significant shifts in the Shannon Diversity index of peaches at every harvest time point across both seasons that samples were collected (p-value < 0.01). Additionally, there was also a significant difference in bacterial diversity for the total bacterial communities versus the viable communities, except for early harvest during the 1st growing season (p-value = 0.02; Fig 5A). Suggesting that not all the bacterial communities present on peaches at harvest are viable, but there is a large portion that are viable and could have a major role in storage and safety.

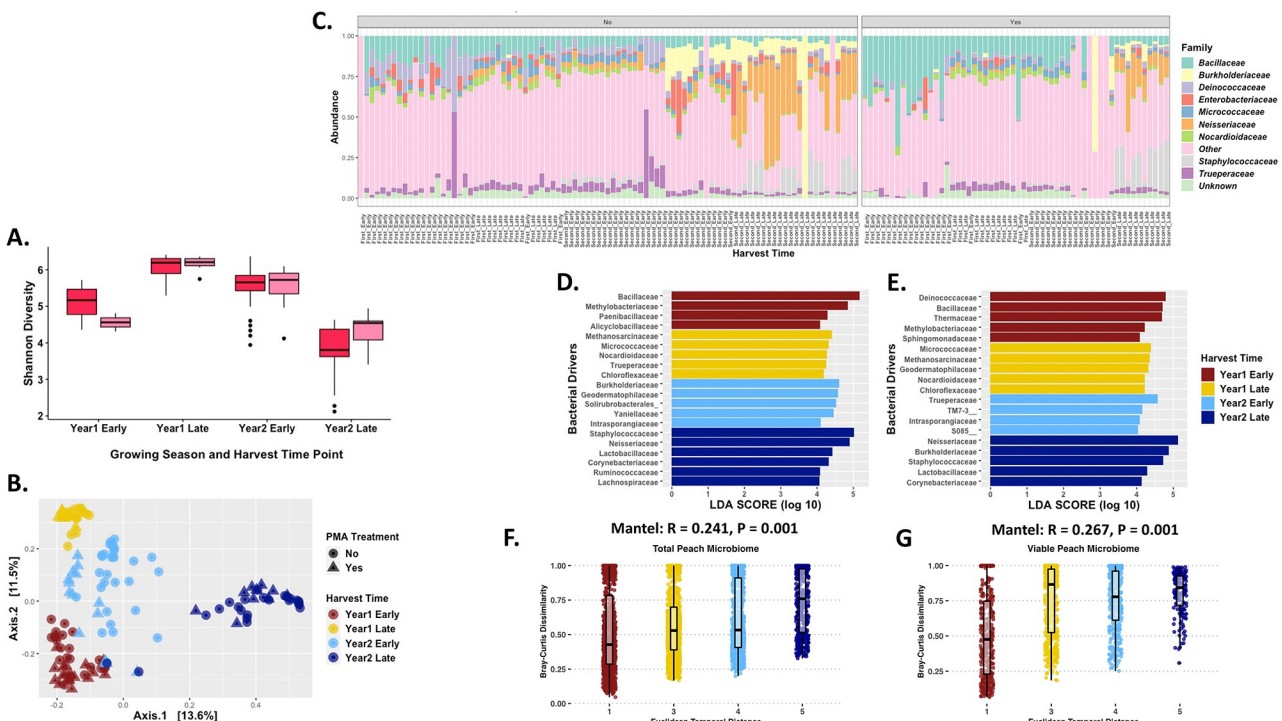

**Fig 5. Bacterial diversity of the peach carposphere.** (A) Shannon Diversity Index for peaches across different time points in the growing season across two growing seasons. Dark pink boxes represent total peach carposphere samples, while the light red boxes are the viable peach carposphere samples. Kruskal-Wallis and Wilcox analysis and showed significance for diversity of peach microbiome over harvesting times (p-value < 0.01). The Kruskal-Wallis p-value was $2.2 \times 10^{-16}$ for the harvesting time periods and p-value of 0.14 for the treatment. Wilcox pairwise analysis among the harvest time periods had significance for different periods. For total early harvest in the first year to total first year late harvest, total second year late harvest, and viable first year late harvest, the p-values were 0.00027, $2.3 \times 10^{-06}$, and $5.8 \times 10^{-06}$, respectively. For total late harvest in the first year to total second year late harvest and viable second year late harvest, the p-values were $2.4 \times 10^{-08}$ and $2.9 \times 10^{-05}$, respectively. For total early harvest in second year to total second year late harvest, viable first year late harvest, and viable second year late harvest, the p-values were $1.7 \times 10^{-08}$, 0.0033, and 0.002, respectively. For total late harvest in the second year to viable first year late harvest and viable second year early harvest, the p-values were $1.6 \times 10^{-07}$ and $5.7 \times 10^{-05}$, respectively. For viable late harvest in the first year to viable second year early harvest and viable second year late harvest, the p-values were 0.0058 and $8.7 \times 10^{-05}$, respectively. Lastly, for viable early harvest in the second year to viable second year late harvest, the p-value was 0.013. (B) PCoA plot based on the Bray-Curtis Dissimilarity Distance Matrix of all the peach carposphere samples. (Permutations = 999; R2 = 0.27; p-value = 0.001). (C) Relative abundance of the top 10 taxa that are present on the different peach carposphere samples, group clustered under "No" represent total microbiome samples that were not treated with PMA, whereas the group clustered under "Yes" represent the viable microbiome samples that were treated with PMA prior to DNA extraction. (D) Linear discriminant analysis Effect Size (LEfSe) analysis of total peach carposphere samples determine those taxa that were most likely driving the differences between the different harvest time points. (E) Linear discriminant analysis Effect Size (LEfSe) analysis of viable peach carposphere samples determine those taxa that were most likely driving the differences between the different harvest time points. (F) Mantel test using the Bray-Curtis Dissimilarity Distance Matrix against the Euclidean temporal distance assesses the temporal correlation of the taxa abundance for total bacterial communities of the peach carposphere. (G) Mantel test using the Bray-Curtis Dissimilarity Distance Matrix against the Euclidean temporal distance assesses the temporal correlation of the taxa abundance for viable bacterial communities of the peach carposphere. Euclidean temporal distance x-axis labels are numerical and represent the harvest time points. In sequential order, 1 is first season early harvest, 2 is first season middle harvest, 3 is first season late harvest, 4 is second season early harvest, and 5 is second season late harvest.

Like the other two fruit types examined in this study, variation in bacterial diversity was further confirmed at the beta diversity level using the Bray-Curtis dissimilarity distance matrix, which found that the samples clustered based on the season and harvest time point in the season (Permanova 999; p-value = 0.001; $R^2$ = 0.27; Fig 5B). Each harvest time point had the total and viable samples cluster together as seen with the other types of fruit, but there was a wider distribution to some of the samples particularly early 2nd year samples. Interestingly, like apples, peach samples from early in both growing seasons clustered closer together than the other harvest time point from the same season, suggesting a potential cyclic nature to the

bacterial communities present on peaches. However, additional work is needed to confirm for sure. Furthermore, the late samples for both growing seasons did not cluster close together, so the communities may start similar but seem to change differently depending on the season and climate during the season.

Examination of the taxonomic abundance of the peach carposphere provides specific bacterial family shifts during the growing season and between seasons. During the 1st growing season there was an increase in the levels of *Micrococcaceae* and *Neisseriaceae* between the early harvest time point and late in the season, which corresponded to a decrease in *Bacillaceae* (Fig 5C). While some of the early 2nd season samples had similar taxonomic profiles to peaches from late in the 1st season, approximately half the samples had *Micrococcaceae* and *Neisseriaceae* mostly replaced by *Burkholderiaceae*. However, by late in the 2nd season those *Burkholderiaceae* were decreasing due to major increases in *Neisseriaceae* and a small increase in *Staphylococcaceae*. The viable bacterial families showed similar patterns with a few exceptions. *Bacillaceae* was prevalent in much higher numbers in the viable microbiome samples, but still decreased between the early and late harvest time periods of the 1st growing season. Additionally, the *Burkholderiaceae* that appeared in the 2nd season samples for the total microbiome were not present or at much lower levels in the viable microbiome samples, suggesting that at least some of those members were dead on the peach carposphere. While there were major shifts in the bacterial diversity during and between growing season on the peach carposphere, there was still a small core microbiome maintained on the fruit. The total core microbiome consisted of five bacterial families, whereas *Bacillaceae* was the only family for the viable microbiome (Table 2). Like apples, this core microbiome was eliminated when including at least 95% of the samples at any abundance.

To further understand the taxa drivers of the bacterial diversity of the peach carposphere at different temporal points in and between growing seasons, a LefSe analysis was conducted. The analysis confirmed some of the taxonomic shifts among the harvest time points in the 1st season, as *Bacillaceae* was the major taxa driver early in the first season, but *Methanosarcinaceae* was a taxa driver late in the 1st season (Fig 5D). The 2nd season analysis confirmed the previous taxonomic profile as *Burkholderiaceae* was the major taxa driver early in the 2nd season, but *Staphylococcaceae* and *Neiseriaceae* were the major taxa drivers by late in the 2nd season. Unlike apples and oranges, there were not major differences in the number of bacterial taxa drivers between total and viable samples for the peach carposphere. However, there were some differences in the specific bacterial families that were the taxa drivers of diversity, for example *Deinococcaceae* was the lead taxa driver in the viable early 1st season population compared to *Bacillaceae*. The late 1st season was similar between total and viable except for *Geodermatophiliaceae* was a taxa driver late in the viable 1st season but not until early in the 2nd season for the total samples. In addition, *Trueperaceae* was the leading viable taxa driver early in the 2nd season, but a taxa driver late in the 1st season for the total samples. *Staphylococcaceae* and *Neisseriaceae* were taxa drivers late in the 2nd season for both sample sets, but *Burkholderiaceae* shifts from early 2nd season in the total samples to late in the 2nd season for viable samples (Fig 5E).

Like apples and oranges, a temporal correlation on the abundance of taxa of the peach carposphere was conducted with Mantel tests on the Bray-Curtis dissimilarity matrix versus the Euclidean temporal distance matrix. The abundance of taxa Mantel tests found a positive correlation and a strong dissimilarity relationship for both total (Fig 5F) and viable (Fig 5G) peach samples. The Sorensen Mantel tests showed community composition of the viable and total peach samples co-varied significantly with temporal distance, which was highlighted by a positive correlation and a strong dissimilarity relationship as samples become more dissimilar at different points of harvest (Fig 6A and 6B).

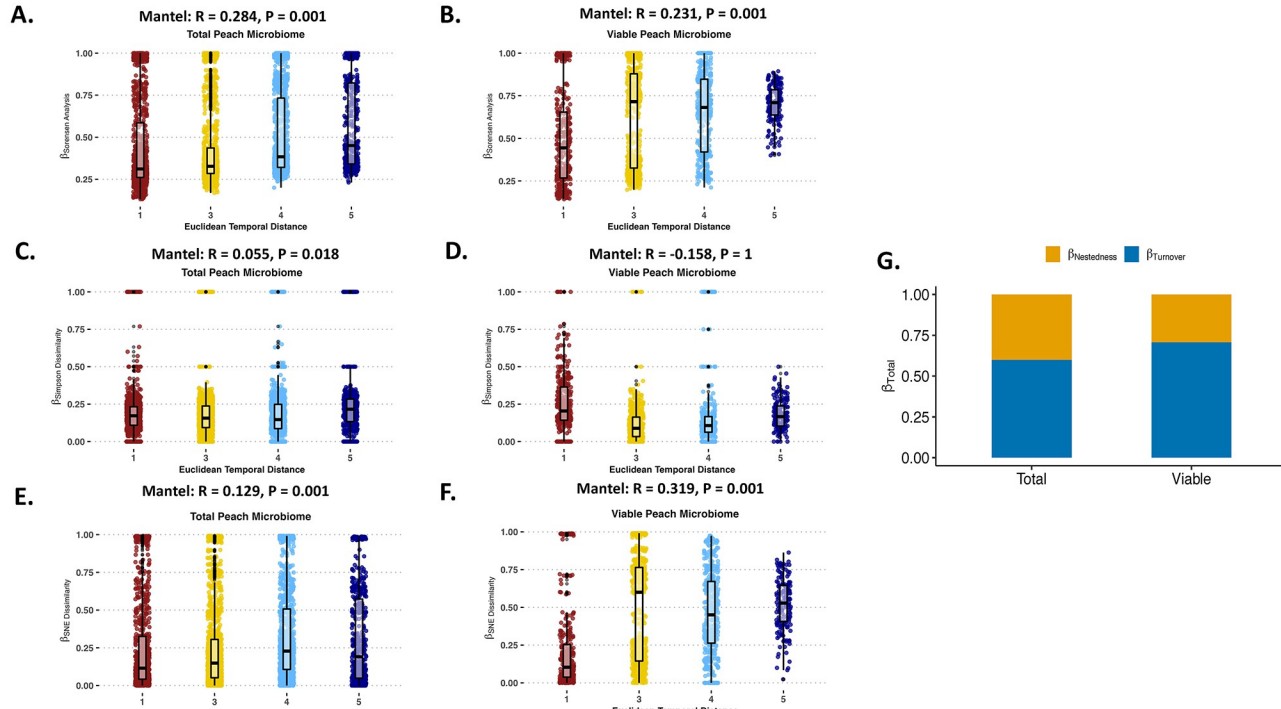

**Fig 6. Role of temporal distance (harvest time point) against bacterial variance for peach carposphere microbiome.** (A) Mantel test using β-Sorensen Pairwise Dissimilarity against the Euclidean temporal distance assesses the temporal correlational role of the taxa presence or absence for total bacterial communities of the peach carposphere. (B) Mantel test using β-Sorensen Pairwise Dissimilarity against the Euclidean temporal distance assesses the temporal correlational role of the taxa presence or absence for viable bacterial communities of the peach carposphere. (C) Mantel test using β-Simpson Pairwise Dissimilarity against the Euclidean temporal distance assesses the temporal correlational role of the species replacement or turnover for total bacterial communities of the peach carposphere. (D) Mantel test using β-Simpson Pairwise Dissimilarity against the Euclidean temporal distance assesses the temporal correlational role of the species replacement or turnover for viable bacterial communities of the peach carposphere. (E) Mantel test using β-SNE Pairwise Dissimilarity (Sorensen—Simpson) against the Euclidean temporal distance assesses the temporal correlational role of the nestedness for total bacterial communities of the peach carposphere. (F) Mantel test using β-SNE Pairwise Dissimilarity (Sorensen—Simpson) against the Euclidean temporal distance assesses the temporal correlational role of the nestedness for viable bacterial communities of the peach carposphere. (G) Compositional variance (β-Total) assesses whether species turnover (β-Simpson) or species nestedness (β-SNE) contributes more to viable and total bacterial communities of the peach carposphere. Euclidean temporal distance x-axis labels are numerical and represent the harvest time points. In sequential order, 1 is first season early harvest, 2 is first season middle harvest, 3 is first season late harvest, 4 is second season early harvest, and 5 is second season late harvest.

The Simpson Mantel tests found a positive correlation and a significant dissimilarity relationship for the total carposphere samples (Fig 6C) but showed a negative correlation and no dissimilarity relationship for viable peach samples over the harvest time points (Fig 6D). Finally, the SNE Mantel tests found community composition of both the total (Fig 6E) and viable (Fig 6F) peach carposphere samples co-varied significantly with temporal distance, which are highlighted by a positive correlation and a strong dissimilarity relationship for both. Unlike apples and oranges, turnover was responsible for at least half the temporal changes on the peach carposphere microbiome for both total (50%) and viable (63%), and thus nestedness had a reduced role in the temporal changes for peaches, total (50%) and viable (37%), which was supported in the LefSe and relative abundance analysis (Fig 6G).

## Discussion

Clearly, though an involved and resource demanding effort, these results are a limited view of tree fruit microbial communities from a limited spectrum of locale, environmental influences,

and variety. This study found that bacterial diversity differed significantly between representatives of the three major types of tree fruit at the point of harvest in commercial orchards located within a limited geographical location (~250 miles apart) in the United States. Although not overly surprising, as differences in the nature of the adjacent land and farm scape features, orchard floor management (bare soil, reflective polymer mulch strips, or vegetated cover), microbiome of the surrounding leaf canopy, and/or the physical surface of the carposphere may collectively have a selective or determinative role in the type of bacteria that are present as transients or long-term colonizer of the fruit prior to harvest. More intensive and extended temporal studies examining these same types of fruit in different locations would be necessary to clarify whether the surfaces of fruit play a role in colonization of different bacteria or if it is due to temporal changes. Interestingly, our study found the bacterial diversity of each type of tree fruit significantly shifted during different harvest time points within and between growing seasons. The magnitude of the bacterial diversity shifts between the fruit harvesting time points varied by the type of fruit, as the apple carposphere did not have any significant shifts compared to peaches and oranges. This could be due to smaller sample size for apples compared to the other two fruit types, sampling points being different for each harvesting period, or variations in the time between sampling points as these were unfortunately not evenly spaced due to logistic challenges. The overall diversity and potential shifts during the growing season could also be related to the chemical composition as well as the physical structure of the carposphere surface, which provide nutrients and environmental protection (e.g., desiccation, UV, and wind) to the carposphere microbiome of each type of fruit that can impact certain bacteria differently throughout the growing season.

Currently, nothing is known about the carposphere microbiome of peaches in the orchards or during fruit development, as this study is the first to characterize the bacterial communities from commercial orchards at the point of harvest. This study found that peaches had the greatest levels of bacterial diversity and temporal variation among the three types of tree fruit, which may be related to the protective indumentum (covering of trichomes or "fine hairs") on the fruit surface. The dense and copious trichome mantle covers a thin cuticular layer containing 15% waxes, 19% cutin, and 63% polysaccharides [42]. The indumentum may provide crucial nutrients and environmental protection to microbial commensals, enabling a more diverse carposphere in peaches than in oranges or apples. Viable bacterial communities on peaches had higher rates of turnover compared to nestedness, suggesting that the majority of the living bacterial community members were leaving and/or dying and were being replaced, instead of a community establishing on the fruit early in development and persisting throughout the growing season. Whereas the total bacterial communities were an even split between turnover and nestedness.

To date, only one study has examined the bacterial diversity of fresh peaches at the point of harvest or later in the processing, and that study used commercial peaches at the point of consumer purchase at the store. While the study did find peaches had higher bacterial diversity compared to the other fruits and vegetables [20], bacterial diversity was reduced at the point of consumer purchase compared to directly off the tree at harvest. This is further supported by the fact that in this study the viable communities were not significantly different from the total bacterial communities on the peach carposphere, meaning most of the bacterial communities were alive and thus capable of dying during storage and/or postharvest processing. The changes/shifts between the orchard in our study and consumers [20] needs further investigation. Our study found five bacterial families in the total core microbiome of peaches including *Bacillaceae* and *Micrococcaceae*, which were both also identified in commercial peaches at the store [20], thus several core bacterial families appear to persist throughout harvesting, processing, and storage to reach consumers.

Apples have a smoother surface compared to peaches and oranges that might not offer as much environmental protection to the bacteria trying to colonize, thus it is not surprising that apples had the lowest level of bacterial diversity among the three fruits in this study. Furthermore, studies have found that the surface wax of apples is composed of a variety of compounds that could negatively influence bacterial colonization such as free fatty acids [42]. Additionally, nutrients present on the apple surface may be in a form that limits the bacterial families with the ability to utilize these sources such as saturated primary alcohols, alkenes, or straight-chain esters [42]. Several studies have examined the apple carposphere microbiome at the point of harvest or under a variety of postharvest conditions [9, 10, 19–21] including postharvest storage [22]. Zhimo et al also recently characterized the carposphere microbiome of apples during fruit development and found a strong temporal shift in the microbiome during development that was driven by turnover [22]. Our study found that apples did have temporal shifts in the bacterial diversity during the growing season that were not significant, which indicates some aspects of the apple carposphere may have an impact on the ability of different bacteria to colonize and survive on the carposphere. Examination of the viable microbiome found it was remarkably similar to the total microbiome on apples, which suggests that most of the bacterial families present on the apple carposphere were alive and adapted to efficiently survive on the surface. Interestingly, our study found the temporal shifts were due to predominately nestedness for total microbiome samples but was an almost even split with turnover for the viable microbiome. Meaning for those viable bacterial communities half the population is colonizing and surviving effectively on the surface, while the other half are constantly being turned over. This is the first study to investigate the temporal dynamics at the point of harvest of these communities throughout and between growing seasons. Additionally, it is also the first to explore the viable apple bacterial communities versus the total apple bacterial communities in a commercial orchard, ultimately only the viable bacterial communities will be important for processing, storage conditions, and produce safety.

Pits in orange surfaces provide a potentially protective area for bacteria to colonize away from various environmental stressors (UV, wind, and other factors). We do know that the chemical composition of the peel contains many aromatics, which are volatile compounds that could potentially impact bacterial colonization. Orange peel oils contain molecules that may affect microbial growth and survival such as monoterpenes, d-limonene, linaleol, and several aldehydes [44, 44]. These essential oils have been explored for antibacterial properties against a few enteric pathogens and some beneficial bacteria. Pathogens were more susceptible to the antibacterial activity of orange oils than beneficial bacteria [46] suggesting that certain members of the orange carposphere microbiome may have a resistance or higher tolerance to these orange peel compounds. To date, the role that essential oils have on the orange carposphere microbiome has not been investigated and will need to be examined to definitively state the oils have a role in the shifts in the bacterial diversity during a growing season. Although further studies are needed to truly understand the role the orange rind's physical and chemical structure have on bacterial colonization.

There has only been a single study examining the bacterial families of citrus carpospheres at the point of harvest or beyond [24]. In that study, clementines and Palmer navel oranges were sampled in packinghouses and during postharvest processing in South Africa [24]. Interestingly, of all the bacterial families identified on citrus in South Africa, only *Methylobacteriaceae* was also identified in our study. This suggests that like studies on apples from around the globe [8], there are significant differences in the bacterial diversity and composition of the citrus carposphere based on geographical location. Our study found oranges have a large core microbiome compared to the other two fruit types throughout the growing season and between seasons, thus suggesting a large/or diverse population can colonize the orange peel.

Like the other tree fruit types in this study, there were significant shifts in the bacterial communities during the growing season and between growing seasons for both the total and viable communities. Although there were no major differences between the viable bacterial population and the total population on oranges. This was further supported by the fact that the temporal changes of the orange carposphere microbiome was due to an even split of nestedness and turnover for these changes for both total and viable communities. Therefore, most of the bacterial communities present on an orange peel at harvest are viable, but who is present changes throughout the growing season and across seasons.

Our study found there were significant changes or shifts in the bacterial diversity of each of the three tree fruit commodities during the growing season and between seasons, but the reason for these temporal changes varied, with nestedness being predominate for apples, turnover for the viable peach microbiome, and an even split for oranges and total peach microbiome. These temporal changes were large enough to result in extremely limited core microbiomes for peaches and apples, particularly the viable samples.

Interestingly, *Methylobacteriaceae* was found in at least 75% of the viable samples for apples and oranges at an abundance of 0.1% (Table 2) and abundance of 0.01% in peaches, and *Methylobacteriaceae* has been found associated with other aspects of the tree fruit orchard for all three types of fruit including the global apple microbiome [8], apples after storage [19], citrus after postharvest treatment [24], as an endophyte of citrus plants [46], and on the bark of peach trees [27]. The genus *Methylobacterium* is considered one of the most prevalent phyllosphere genera that is present on almost every plant [47, 49], and has a wide range of vital roles in plant physiology from growth stimulation via hormone secretion [50] to heavy metal sequestering [51]. Recently, it was demonstrated on tree leaves that there are diverse and dynamic *Methylobacterium* communities on the phyllosphere that shift over short temporal ranges, which is suggested to be climatic adaption to seasonal variation [52]. Our results that the bacterial family is quite frequently present on the carposphere, suggests it might have a role for tree fruit producers for growing, postharvest processing, and storage. Nevertheless, the dynamic nature seen in leaves and this study would suggest specific species may vary during the growing season and between seasons. Therefore, the specific role *Methylobacteriaceae* has in tree fruit production needs further assessment.

The major limitation of this study was the limited access to numerous commercial orchards, as each commercial orchard requires the trust and collaboration of that grower, and therefore building these relationships requires a large amount of work and limits the number of commercial orchards and the geographical locations. Additionally, as these were fully functional commercial orchards there were limitations of access to collect fruits that did not interfere with their operations, thus it resulted in an uneven collection of the different types of tree fruit during the two growing seasons. With these limitations in mind, a future study could look to examine different geographical regions around the United States and/or the world to assess temporal changes on more of a global scale. This study lays the foundation for several future studies, including (1) the role the viable core microbiome at the point of harvest has in tree fruit storage, (2) how do the bacterial families that shift during the growing season impact post-harvest processing, and (3) further development of a tool for the verification and validation of wash water systems for food safety and quality.

## Conclusions

Overall, this is the first study to investigate the temporal changes of the bacterial communities on three major tree fruit commodities collected directly in commercial orchards. This was also the first study to compare the viable bacterial communities of the tree fruit to the total bacterial

communities (living vs dead) present on the carposphere. Finally, this study is also the first to characterize either the viable or total bacterial communities of both the peach and orange carpospheres directly from the orchard. This study provides a strong, albeit narrow, foundation to understand the bacterial communities present on three major tree fruit carpospheres at the point of harvest and will provide the tree fruit industry with vital data for improving postharvest processing and fruit safety in the future.

## Supporting information

**S1 Fig. Comparison between 515F and 799F primers by fruit type.** Sequences were rarefied by random permutation to 2000 sequences per sample. Note that colors should only be compared within a fruit type. Unidentified bacteria were classified only to the Kingdom level by RDP. Low abundant bacteria are bacteria taxa below the top 10 most abundant taxa at the family level, for color reasons.
(TIF)

**S2 Fig. Different alpha diversity metrics for total microbiome.** Comparing the total bacterial communities of the carposphere of the tree types of the tree fruit used in this study with different alpha diversity metrics.
(TIF)

**S3 Fig. Different alpha diversity metrics for viable microbiome.** Comparing the viable bacterial communities of the carposphere of the tree types of the tree fruit used in this study with different alpha diversity metrics.
(TIF)

**S4 Fig. Beta diversity of total and viable tree fruit microbiome.** (A) Beta diversity of all the total carposphere samples for the three types of tree fruit used in this study. (B) Beta diversity of all the viable carposphere samples for the three types of tree fruit used in this study.
(TIF)

**S5 Fig. Abundance heatmap of top 14 bacterial families for the three tree fruit types.** Heat map of the abundance of the top 14 bacterial communities for the three tree fruit types, further split by PMA treatment. As color darkens, the more abundant that bacterial family is in the sample. Detection was at 0.1% and prevalence of 75% were used, same as the core microbiome analysis. All samples are represented as total (non-PMA treated) and viable (PMA treated) for the respective fruit. The color scale represents the abundance of each of the top 14 bacteria present for each sample collected, going from light green to dark green.
(TIF)

## Acknowledgments

The authors thank Chris Chabot for technical assistance throughout the study, and Cristina Alcaraz for assistance with many different aspects of the project. Additionally, the authors thank the members of the tree fruit industry that collaborated on this study by providing access to their commercial orchards on numerous occasions to allow for the collection of the different types of tree fruit.

## Author Contributions

**Conceptualization:** Madison Goforth, Margarethe A. Cooper, Trevor V. Suslow, Gilberto E. Flores, Craig T. Parker, Rachel Mackelprang, Kerry K. Cooper.

**Formal analysis:** Madison Goforth, Andrew S. Oliver, Gilberto E. Flores, Craig T. Parker, Kerry K. Cooper.

**Funding acquisition:** Trevor V. Suslow, Kerry K. Cooper.

**Methodology:** Madison Goforth, Margarethe A. Cooper, Janneth Pinzon, Mariya Skots, Victoria Obergh, Trevor V. Suslow, Gilberto E. Flores, Steven Huynh, Craig T. Parker, Kerry K. Cooper.

**Project administration:** Kerry K. Cooper.

**Supervision:** Kerry K. Cooper.

**Visualization:** Kerry K. Cooper.

**Writing – original draft:** Madison Goforth, Margarethe A. Cooper, Andrew S. Oliver, Gilberto E. Flores, Craig T. Parker, Kerry K. Cooper.

**Writing – review & editing:** Madison Goforth, Margarethe A. Cooper, Andrew S. Oliver, Trevor V. Suslow, Gilberto E. Flores, Craig T. Parker, Rachel Mackelprang, Kerry K. Cooper.

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
