## [Decision Letter · Decision Letter 0]

31 Oct 2023

PONE-D-23-29402Bacterial community shifts of commercial apples, oranges, and peaches at different harvest points across multiple growing seasonsPLOS ONE

Dear Dr. Cooper,

Thank you for submitting your manuscript to PLOS ONE. After careful consideration, we feel that it has merit but does not fully meet PLOS ONE’s publication criteria as it currently stands. Therefore, we invite you to submit a revised version of the manuscript that addresses the points raised during the review process.

We look forward to receiving your revised manuscript.

Kind regards,

Massimiliano Cardinale, PhD

Academic Editor

PLOS ONE

Journal Requirements:

The authors thank Chris Chabot for technical assistance throughout the study, and Cristina Alcaraz for assistance with many different aspects of the project. Funding for the study was provided by the United States Department of Agriculture (USDA), National Institute of Food and Agriculture (NIFA) Award #2017-67018-26173 awarded to Kerry Cooper and Trevor Suslow, and Technology and Research Initiative Fund (TRIF) provided to Kerry Cooper by the University of Arizona. No funding agency had any role in the study design, data collection and analysis, decision to publish, or preparation of the manuscript.

Funding for the study was provided by the United States Department of Agriculture (USDA), National Institute of Food and Agriculture (NIFA) Award #2017-67018-26173 awarded to Kerry Cooper and Trevor Suslow, and Technology and Research Initiative Fund (TRIF) provided to Kerry Cooper by the University of Arizona. No funding agency had any role in the study design, data collection and analysis, decision to publish, or preparation of the manuscript. USDA NIFA (https://www.nifa.usda.gov/) and UArizona TRIF (https://research.arizona.edu/trif#:~:text=Through%20TRIF%2C%20or%20the%20Technology,largest%20economic%20engines%20for%20Arizona.)

6. Please amend the manuscript submission data (via Edit Submission) to include author Dr. Victoria Obergh.

7. We note that you have included the phrase “data not shown” in your manuscript. Unfortunately, this does not meet our data sharing requirements. PLOS does not permit references to inaccessible data. We require that authors provide all relevant data within the paper, Supporting Information files, or in an acceptable, public repository. Please add a citation to support this phrase or upload the data that corresponds with these findings to a stable repository (such as Figshare or Dryad) and provide and URLs, DOIs, or accession numbers that may be used to access these data. Or, if the data are not a core part of the research being presented in your study, we ask that you remove the phrase that refers to these data.

Reviewers' comments:

Reviewer's Responses to Questions

**Comments to the Author**

1. Is the manuscript technically sound, and do the data support the conclusions?

Reviewer #1: Yes

Reviewer #2: Yes

2. Has the statistical analysis been performed appropriately and rigorously? 

Reviewer #1: Yes

Reviewer #2: Yes

3. Have the authors made all data underlying the findings in their manuscript fully available?

Reviewer #1: Yes

Reviewer #2: Yes

4. Is the manuscript presented in an intelligible fashion and written in standard English?

Reviewer #1: Yes

Reviewer #2: Yes

5. Review Comments to the Author

Reviewer #1: The study, titled "Bacterial Community Shifts of Commercial Apples, Oranges, and Peaches at Different Harvest Points Across Multiple Growing Seasons," sets a clear focus on understanding the shifts in bacterial communities associated with commercially significant fruits. The primary goal of this research was to establish a foundational profile of bacterial communities present on apples, peaches, and Navel oranges at the moment of harvest. This is a critical juncture in the life cycle of these fruits, as it marks the point at which they transition from the tree to the postharvest handling and processing stages. The study's findings hold substantial importance as they offer a comprehensive view of the temporal dynamics of bacterial communities on major commercial tree fruits. This temporal aspect adds depth to our understanding of the microbial ecology associated with these fruits. By examining the bacterial communities at harvest, the study provides valuable insights into how these communities might influence the postharvest packing and processing stages. Understanding these dynamics is crucial for ensuring the quality and safety of the fruits as they move through the supply chain.

The data is presented with meticulous attention to detail, demonstrating a thorough and comprehensive approach to the research. This not only adds credibility to the findings but also enhances the overall quality of the paper. The clarity and coherence of the writing contribute significantly to the paper's overall impact. The study makes a noteworthy contribution to the existing literature in this field. By providing a detailed analysis of bacterial communities in commercially relevant fruits, it fills a crucial gap in our understanding of fruit microbiology.

Based on the quality of research, the thoroughness of data presentation, and the significance of the findings, I believe this study holds substantial merit and should certainly be considered for publication. It not only advances our knowledge in this specific area of research but also has broader implications for the fruit industry and food safety practices.

Reviewer #2: The manuscript describes a study that examines the bacterial communities of apples, oranges, and peaches at different harvest points across multiple growing seasons. The work appears to be well-designed and the results are presented in a clear and organized manner. The findings of the study have the potential to contribute to our understanding of the bacterial communities associated with these fruits and their changes over time.

1. The abstract is well-written and concise, but it could be more informative and specific about the main findings and implications of the study. For example, you could mention the names of the dominant bacterial taxa or families in each fruit type.

2. The introduction could be more concise and focused on the main research question and objectives.

3. The introduction should clearly state the problem or question being addressed in the study. The authors have mentioned that they wanted to explore the bacterial communities of apples, oranges, and peaches at different harvest points. However, a more specific statement of the problem, such as "How do the bacterial communities of these fruits change over time?", would be more engaging and would provide a clearer focus for the reader.

4. I am confused by the sample size mentioned in the manuscript. According to the author's description, 50 fruits were collected from each tree, and 10 or 5 fruits were combined into one composite sample. How many trees were collected in total? How many samples in each composite sample? The author provided the number of sequenced samples in Table 1, but it does not match the number shown in Supplemental Figure 4. Therefore, the author needs to clarify the process of determining the sample size in the Methods section.

5. L167: What do you mean by commercial orchards? L167-168: Why did you choose these orchards? Any rules in choosing the geographical location?

6. The discussion should acknowledge any limitations of the study, and provide some suggestions for future research in the field.

6. PLOS authors have the option to publish the peer review history of their article (what does this mean?). If published, this will include your full peer review and any attached files.

Reviewer #1: **Yes: **Walid Ellouze

Reviewer #2: No

---

## [Author Response · Author response to Decision Letter 0]

30 Dec 2023

Response to reviewers:

We thank the reviewers for their kind comments and excellent insights into improving our manuscript prior to publication. We have addressed all the comments in the revised manuscript and the changes for each specific comment are highlighted below.

Reviewer #1: The study, titled "Bacterial Community Shifts of Commercial Apples, Oranges, and Peaches at Different Harvest Points Across Multiple Growing Seasons," sets a clear focus on understanding the shifts in bacterial communities associated with commercially significant fruits. The primary goal of this research was to establish a foundational profile of bacterial communities present on apples, peaches, and Navel oranges at the moment of harvest. This is a critical juncture in the life cycle of these fruits, as it marks the point at which they transition from the tree to the postharvest handling and processing stages. The study's findings hold substantial importance as they offer a comprehensive view of the temporal dynamics of bacterial communities on major commercial tree fruits. This temporal aspect adds depth to our understanding of the microbial ecology associated with these fruits. By examining the bacterial communities at harvest, the study provides valuable insights into how these communities might influence the postharvest packing and processing stages. Understanding these dynamics is crucial for ensuring the quality and safety of the fruits as they move through the supply chain.

The data is presented with meticulous attention to detail, demonstrating a thorough and comprehensive approach to the research. This not only adds credibility to the findings but also enhances the overall quality of the paper. The clarity and coherence of the writing contribute significantly to the paper's overall impact. The study makes a noteworthy contribution to the existing literature in this field. By providing a detailed analysis of bacterial communities in commercially relevant fruits, it fills a crucial gap in our understanding of fruit microbiology. Based on the quality of research, the thoroughness of data presentation, and the significance of the findings, I believe this study holds substantial merit and should certainly be considered for publication. It not only advances our knowledge in this specific area of research but also has broader implications for the fruit industry and food safety practices.

We thank Reviewer 1 for their very kind words and appreciation of the amount of hard work that went into the study, and that they see substantial merit to the research and the publication. We agree that it will have a strong impact on the postharvest packing and processing for the tree fruit industry. 

Reviewer #2: The manuscript describes a study that examines the bacterial communities of apples, oranges, and peaches at different harvest points across multiple growing seasons. The work appears to be well-designed and the results are presented in a clear and organized manner. The findings of the study have the potential to contribute to our understanding of the bacterial communities associated with these fruits and their changes over time.

The authors also thank Reviewer 2 for their positive feedback on the study and publication, we really appreciate the guidance to making the manuscript stronger and have highlighted the changes made to each of the comments below.

1. The abstract is well-written and concise, but it could be more informative and specific about the main findings and implications of the study. For example, you could mention the names of the dominant bacterial taxa or families in each fruit type.

We thank the reviewer for suggesting the addition of more specific findings to improve the abstract. We have added the core microbiome bacterial families found on each type of tree fruit to the abstract and what this means to the results of the study.

2. The introduction could be more concise and focused on the main research question and objectives.

We have made the introduction more concise, eliminating several points that do not directly focus ont eh main research question of the study. Therefore, the introduction is overall now more focused on setting up for our specific objectives of the study.

3. The introduction should clearly state the problem or question being addressed in the study. The authors have mentioned that they wanted to explore the bacterial communities of apples, oranges, and peaches at different harvest points. However, a more specific statement of the problem, such as "How do the bacterial communities of these fruits change over time?", would be more engaging and would provide a clearer focus for the reader.

We have added the specific problem that the study was looking to address to the introduction as recommended by the reviewer.

4. I am confused by the sample size mentioned in the manuscript. According to the author's description, 50 fruits were collected from each tree, and 10 or 5 fruits were combined into one composite sample. How many trees were collected in total? How many samples in each composite sample? The author provided the number of sequenced samples in Table 1, but it does not match the number shown in Supplemental Figure 4. Therefore, the author needs to clarify the process of determining the sample size in the Methods section.

We understand the confusion of the reviewer, and have changed the wording of the sample collection section of the materials and methods section to make it clearer to the reader. We have addressed all the concerns including how many trees were collected and how many samples in a composite sample. The difference in the numbers of Table 1 and the Supplemental Figure 4 is due to the elimination of some samples during rarefication to 2000 reads, which has been clarified in the methods section too.

5. L167: What do you mean by commercial orchards? L167-168: Why did you choose these orchards? Any rules in choosing the geographical location?

We have added a statement about the definition of commercial orchard to the methods section, and also a statement about why these orchards were chosen for the study. 

6. The discussion should acknowledge any limitations of the study, and provide some suggestions for future research in the field.

We have added a paragraph at the end of the discussion that addresses the limitations of the study and the future work based on our study.

---

## [Decision Letter · Decision Letter 1]

4 Jan 2024

Bacterial community shifts of commercial apples, oranges, and peaches at different harvest points across multiple growing seasons

PONE-D-23-29402R1

Dear Dr. Cooper,

We’re pleased to inform you that your manuscript has been judged scientifically suitable for publication and will be formally accepted for publication once it meets all outstanding technical requirements.

Kind regards,

Massimiliano Cardinale, PhD

Academic Editor

PLOS ONE

Additional Editor Comments (optional):

Reviewers' comments:

Reviewer's Responses to Questions

**Comments to the Author**

1. If the authors have adequately addressed your comments raised in a previous round of review and you feel that this manuscript is now acceptable for publication, you may indicate that here to bypass the “Comments to the Author” section, enter your conflict of interest statement in the “Confidential to Editor” section, and submit your "Accept" recommendation.

Reviewer #2: All comments have been addressed

2. Is the manuscript technically sound, and do the data support the conclusions?

Reviewer #2: Yes

3. Has the statistical analysis been performed appropriately and rigorously? 

Reviewer #2: Yes

4. Have the authors made all data underlying the findings in their manuscript fully available?

Reviewer #2: Yes

5. Is the manuscript presented in an intelligible fashion and written in standard English?

Reviewer #2: Yes

6. Review Comments to the Author

Reviewer #2: I appreciate their efforts to address the comments and suggestions from the previous round of review. I have no further comments or questions.

7. PLOS authors have the option to publish the peer review history of their article (what does this mean?). If published, this will include your full peer review and any attached files.

Reviewer #2: No

---

## [Editor Report · Acceptance letter]

23 Feb 2024

PONE-D-23-29402R1 

PLOS ONE

Dear Dr. Cooper, 

I'm pleased to inform you that your manuscript has been deemed suitable for publication in PLOS ONE. Congratulations! Your manuscript is now being handed over to our production team.

Kind regards, 

on behalf of

Dr. Massimiliano Cardinale 

Academic Editor

PLOS ONE